# Different Preharvest Diseases in Garlic and Their Eco-Friendly Management Strategies

**DOI:** 10.3390/plants13020267

**Published:** 2024-01-17

**Authors:** Hadiqa Anum, Yuxin Tong, Ruifeng Cheng

**Affiliations:** 1Institute of Environment and Sustainable Development in Agriculture, Chinese Academy of Agricultural Sciences, Beijing 100081, China; hadiqaanum120@gmail.com (H.A.); chengruifeng@caas.cn (R.C.); 2Key Laboratory of Energy Conservation and Waste Management of Agricultural Structures, Ministry of Agriculture, Beijing 100081, China

**Keywords:** *Allium*, disease prevention, nematodes, plant pathogens, tissues culture, yield optimization

## Abstract

Background: garlic reproduces mainly through clove planting, as sexual reproduction via seeds is uncommon. Growers encounter challenges with pathogens due to the larger size and vegetative nature of seed cloves, as well as the storage conditions conducive to fungal growth. Some Phyto-pathogenic fungi, previously unrecognized as garlic infections, can remain latent within bulb tissues long after harvest. Although outwardly healthy, these infected bulbs may develop rot under specific conditions. Aim of review: planting diseased seed cloves can contaminate field soil, with some fungal and bacterial infections persisting for extended periods. The substantial size of seed cloves makes complete eradication of deeply ingrained infections difficult, despite the use of systemic fungicides during the preplanting and postharvest phases. Additionally, viruses, resistant to fungicides, persist in vegetative material. They are prevalent in much of the garlic used for planting, and their host vectors are difficult to eliminate. To address these challenges, tissue-culture techniques are increasingly employed to produce disease-free planting stock. Key scientific concepts of the review: garlic faces a concealed spectrum of diseases that pose a global challenge, encompassing fungal threats like Fusarium’s vascular wilt and Alternaria’s moldy rot, bacterial blights, and the elusive garlic yellow stripe virus. The struggle to eliminate deeply ingrained infections is exacerbated by the substantial size of seed cloves. Moreover, viruses persist in garlic seeds, spreading through carrier vectors, and remain unaffected by fungicides. This review emphasizes eco-friendly strategies to address these challenges, focusing on preventive measures, biocontrol agents, and plant extracts. Tissue-culture techniques emerge as a promising solution for generating disease-free garlic planting material. The review advocates for ongoing research to ensure sustainable garlic cultivation, recognizing the imperative of safeguarding this culinary staple from an array of fungal and viral threats.

## 1. Introduction

The cycle of garlic (*Allium sativum* L.) production depends heavily on seed-garlic germination. For optimum crop development, production, and yield, high-quality seed stock is necessary. Growers may build a solid foundation for productive garlic growing by starting with healthy and disease-free garlic seeds. Garlic holds a global agricultural reign with an annual production exceeding 28 million tonnes in 2021 [1]. China, the undisputed king, dominates the market, contributing a staggering 73% of this total, equivalent to filling nearly 11.3 million Olympic-sized swimming pools with garlic cloves [2] (Table 1). The production and financial success of garlic farms are directly impacted by the quality of the seed garlic. Breeders value the ability of some cultivars of this plant to reproduce from genuine seed [3]. U.S. Patent 5746024 covers the method of making genuine seeds from garlic [4]. The utilization of clonal propagules, such as seed cloves [5] or, in rare circumstances, inflorescence bulbils [6], is the most common technique for growing garlic. Our primary concern in this review is the control of infections that harm seed cloves or are spread by them.

Garlic cloves are typically obtained from plants that have been harvested in the late summer (e.g., in the Pacific Northwest region of the United States, from early July to mid-September) and then stored for planting in the autumn (mid-September to November) or, for some varieties, in the early spring (March) [13,14]. In order to prevent sprouting during storage and ensure quick germination and development after planting, storage conditions with a temperature range of around 12–14 °C and a relative humidity of 55% are used [15]. Since storing garlic bulbs at temperatures close to 5 °C for lengthy periods of time increases the danger of sprouting, or even a storage temperature of roughly 10 °C [16], garlic cultivars intended for spring planting have been effectively preserved at 0 °C or −3 °C, even though some varieties can be susceptible to such low temperatures [17,18].

## 2. Garlic Seed Cloves as Habitat Pathogens

In order to maintain the development of healthy seed stock, effective disease management is essential in the seed-garlic industry. Garlic plants are susceptible to diseases, which can lead to decreased yields, poor bulb quality, and greater susceptibility to secondary infections. Growers may lessen the effects of infections and preserve the health and vigor of the seed-garlic crop by implementing good disease-control practices into action. Implementing disease-control strategies in garlic production systems disrupts the perpetuation of pathogens across subsequent harvests, thereby safeguarding the long-term sustainability and productivity of the crop. Garlic cloves are larger than the true seeds of the majority of other field crops, and because they are stored at temperatures between 10 °C and 14 °C, disease control can be challenging. As a result of their size, even systemic fungicide treatment during preplanting or postharvest dips is insufficient to entirely eliminate deeply entrenched diseases [19]. The application of fungicidal dips therefore does not ensure a considerable reduction in inoculum inside the planting stock, unlike with the majority of genuine seeds. Although systemic fungicides are authorized for use against *Fusarium* and *Penicillium*-caused bulb rots in ornamental flowers as well as *Allium* species, fungicidal dips target recently established infections that are superficial [20,21]. Additionally, infestations of the wheat curl mite (*Eriophyes tulipae* Keifer) and bulb mites (*Tyrophagus* spp. and *Rhizoglyphus* spp.) during storage might promote the development of garlic rot. According to [4], the same storage conditions that encourage fungal development also support mite populations. *Histiostoma onioni* Eraky, *Rhizoglyphus robini* Claparède, and *Tyrophagus putrescentiae* Shrank have been demonstrated as capable vectors of *A. ochraceus* Wilhelm, *Aspergillus niger* Van Tieghem, *Gibberella fujikuroi* (Sawada) Ito, *Penicillium* spp., and other fungi [22]. Despite relying on chemical agents, controlling mites in garlic remains a challenge. There is a detailed manual on garlic diseases and pests [23] (Figure 1).

### 2.1. Fungal Pathogens

The production of garlic seed is vulnerable to fungal infections. Multiple diseases, including bulb rots and leaf spots, can be brought on by pathogens, including *Fusarium*, *Penicillium*, and *Alternaria* species. These microorganisms can remain latent and grow in storage environments, infecting garlic bulbs either before planting or while being stored. Effective disease-management techniques depend on having a thorough understanding of the symptoms and harm that fungi-based infections produce. *Fusarium oxysporum* Schlechtend.:Fr. f. sp. cepae H.N. Hans. W.C. Snyder and H.N. Hans. (Foc), *Alternaria embellisia* (syn. *Embellisia allii*), *Aspergillus ochraceus*, *Aspergillus niger, Penicillium hirsutum* Dierckx, and *F. proliferatum* (Matsushima) Nirenberg are the main fungi that attack garlic bulbs while they are being stored [24]. 

There have also been reports of *Fusarium verticillioides* (Sacc.) Nirenberg and *Botrytis porri* Buchw. causing rot [24]. It is important to remember, though, that not every isolate with these names necessarily displays aggressive pathogenic behavior. Among pathogenic species, isolates of *A. ochraceus, A. niger, and E. allii* exhibit varying degrees of virulence. Notably, *E. allii* presents a greater threat under damp field conditions [20]. Furthermore, it appears that some isolates of the highly aggressive *Fusarium oxysporum* f. sp. cepae and *F. proliferatum*, which are also found in onions, are less aggressive in garlic, especially after the garlic bulbs have undergone postharvest aging or “hardening” [25]. It is still completely unclear how long some of these infections may remain latent or quiescent in tissues, as was the case with Velásquez-Valle et al. [26], who comment on *Fusarium culmorum* (Wm. G. Sm.) Sacc.

Dugan, Hellier, and Lupien [24] identified three or more of these pathogenic species from each of the seven lots examined, including six lots from diverse locations in the USA and one lot from mainland China, in a study of commercially available asymptomatic seed garlic. Buslyk, et al. [27] reported molecular–genetic techniques for the identification and classification of mycotoxin-producing fungus found in garlic. The *Penicillium* species, which has recently been the focus of debates surrounding its proper name, is one of the most active fungal infections responsible for the rot in garlic. Prior to its synonym, *P. hirsutum*, being more often used, the term *P. corymbiferum* (*P. verrucosum* var. corymbiferum (Westling) Samson, Stolk, and Hadlok), was also frequently used [28]. *Penicillium* species that cause rot in garlic have been referred to by a variety of names [29]. Only isolates identified as *Penicillium allii* Vincent and Pitt were shown to be extremely harmful to garlic by Salinas and Cavagnaro [30], whereas isolates classified as *P. hirsutum* were less aggressive.

Gálvez and Palmero [31] used the name *P. allii* to refer to pathogenic isolates, while [28] utilized the name *P. hirsutum*. Despite acknowledging diverse preferences among isolates, Dugan [4] opted for a broad application of the name *P. hirsutum* to all garlic-pathogenic isolates in his study, motivated by the presence of a characteristically dark exudate in some isolates [32]. Prior to the discovery of *P. allii*, several species in the Corymbifera section, notably *P. hirsutum* var. allii (Vincent and Pitt) Fisvad [33], were thought to be variations of *P. hirsutum*. While the type material for *P. hirsutum* is a neotype isolated from aphids, the viable-type material (strictly speaking, extype) for *P. allii* comes from garlic [34]. *Sclerotium cepivorum* Berk., the causative agent of white rot, poses a grave threat to garlic agriculture, inflicting substantial yield losses and persistent soil contamination. Recognizing the significant economic and ecological consequences, some regions have implemented stringent quarantine protocols for seed cloves. These measures are aimed at curtailing the dissemination of the pathogen, safeguarding garlic crops, and ensuring agricultural sustainability. This proactive approach to disease management shields garlic-farming communities from the devastating effects of white rot. [35]. Although total eradication may not be possible, hot water treatments can be successful in controlling *S. cepivorum* in planting stock [36,37] (Table 2).

### 2.2. Fungal Pathogens in Soil

A pathogen that also affects onions, *Fusarium oxysporum* f. sp. cepae, forms chlamydospores and has the capacity for long-term survival [51]. Unlike *F. proliferatum*, which does not form chlamydospores [52], it may survive in soil for a long time if it is present in agricultural wastes [53]. In simulated winter conditions in field soil, [24] found that both *F. proliferatum* and *F. oxysporum* survived prolonged freezing. The potential of *Fusarium verticillioides* to thrive in agricultural settings is also well documented [54]. *A. ochraceus* and *Botrytis porri* both have the ability to create sclerotia, and both can do so in a sizeable amount [55]. *Sclerotium cepivorum*, as its name implies, also generates sclerotia, which may survive for years without a host plant [56]. To reduce the occurrence of *S. cepivorum* sclerotia, many management techniques have been devised, such as the use of substances that stimulate germination by imitating *Allium* root exudates [57]. *Penicillium hirsutum*, on the other hand, does not show long-term persistence in the soil [58] (Figure 2).

### 2.3. Mycotoxin

While the emphasis of this analysis is focused on seed garlic, it is important to recognize that a number of fungi have the capacity to create poisons that might be substantial in table garlic. *F. proliferatum*, which was first discovered in market garlic in Germany by [59], has now been found in North American garlic fields [60]. Despite the fact that *F. verticillioides* was just recently identified as a pathogen that causes garlic to rot [4], the organism has been studied for its capacity to produce mycotoxin [20]. Fumonisins are most frequently produced by *F. proliferatum* and *F. verticillioides*; *F. proliferatum* also produces other mycotoxins [61].

## 3. Bacteria and Virus

A garlic disease known as “maladie café au lait,” attributed to *Pseudomonas fluorescens* and caused by the pathogen *Migula*, poses a significant threat to garlic cultivation, impacting both yield and quality [62]. On at least one pest list for garlic, *Burkholderia cepacia* (Palleroni and Holmes) is designated as a controlled organism [63]. Several organisms, including *E. chrysanthemi*, *Erwinia carotovora*, *Enterobacter cloacae*, *Pseudomonas gladioli*, and *Burkholder*, can cause soft rot in onions and garlic. However, these organisms are more damaging to onions than they are to garlic [64] (Table 3 and Table 4).

Many studies, most notably [87], have documented yield losses brought on by viral infections, particularly the prevalent OYDV. Numerous viruses can coexist in a single illness, which is known as a mixed infection [88]. Several mite-borne viruses that infect *Allium* species are included in the relatively new genus *Allexivirus* in the family *Flexiviridae*, in addition to the viruses already described. These include the garlic mite-borne filamentous virus (GarMbFV), garlic virus A (Gar V-A), garlic virus B (Gar V-B), garlic virus C (Gar V-C), garlic virus D (Gar V-D), garlic virus E (Gar V-E), and garlic virus X (Gar V-X) [89].

## 4. Nematodes

Nematode infections, in particular *Ditylenchus dipsaci*, pose considerable problems to the growth of seed garlic. These tiny roundworms invade the roots, bulbs, and leaves of the garlic plant, resulting in stunted development, deformed bulbs, and decreased harvests. Seed garlic contaminated with nematodes can hasten the spread of nematodes in succeeding harvests, escalating the harm. For a healthy seed supply to be maintained, effective nematode management is essential. Several *Allium* species in temperate climates are negatively impacted by the pest *Ditylenchus dipsaci* (Kühn) Filipjev [90,91,92]. Nematode numbers in garlic seeds have been managed by hot water treatments. This method efficiently kills nematodes without doing much harm to the garlic by dipping seed cloves or bulbs in hot water for certain amounts of time and temperature. Nematode populations can be reduced by hot water treatments, which also help to stop the transmission of nematode-borne illnesses to future crops. The nematode can be controlled with hot water treatments, especially when they are followed by a long soak in cold water [93,94]. The adaptability of different garlic cultivars to greater temperatures without suffering damage varies; thus, care should be taken. Finding temperatures that are high enough to kill the worm without seriously harming the garlic plants is essential. Conducting lengthy investigations may be difficult due to the restricted availability of germplasm in significant numbers for testing. However, hot water treatments can be helpful, when there is a plentiful supply of a specific garlic species (Table 5).

## 5. Disease-Free Planting Stock

Garlic that is free of disease may now be produced by utilizing tissue-culture methods, particularly a meristem culture employing one or two leaf primordia. Researchers can get rid of viruses and grow clean, virus-free, garlic plants by cultivating tiny pieces of plant tissue in a controlled lab setting. Tissue culture makes it possible to quickly multiply disease-free planting stock, enabling the production of healthy seed garlic. Since garlic cloves are vegetative propagules, viruses continue to infect garlic plants, whether they are cultivated for the table or as seed stock. Although most seed garlic contains viruses, not all of them are obviously harmful [103]. Alternative management approaches are preferred since the chemical treatment of numerous viral vectors (aphids, nematodes, and thrips) is challenging and expensive [104] (Figure 3).

The creation of virus-free garlic plants via tissue culture, specifically meristem culture using one or two leaf primordia, has become possible. According to study [105], clones grown by tissue cultures that are virus-free have better yields and profitability than clones that are infected. Improvements in methods, such as the use of inflorescence bulbil primordia, have enhanced success rates even if the successive cultivation of meristems from shoot tips does not always ensure total viral eradication. In vegetatively grown *Allium cepa* var. ascalonicum (shallots), viral infections have been eradicated using comparable tissue-culture techniques [106]. Garlic has been successfully cryopreserved, and the stocks are frequently virus-free. Nevertheless, the efficiency of cryopreservation may differ depending on the tissues (bulbils vs. cloves) and varieties of garlic (hard-neck versus soft-neck). In places like California, Australia, and Canada [107], tissue-culture techniques are currently used to produce disease-free commercial planting stock. It is crucial to keep in mind that producers who engage in large-scale production may most easily afford this equipment.

### 5.1. Resistant Cultivars

The management of viral and fungal infections in garlic has been investigated in regard to prospective techniques, including resistance and/or tolerance. Observing the absence of viral symptoms or the inability to detect viruses in particular cultivars has been the focus of some reports (e.g., [3,105]), whereas other studies have gone through extensive testing to determine garlic’s resistance to the viruses leek yellow stripe virus (LYSV) and onion yellow dwarf virus (OYDV) [108]. There are also reports describing resistance to a number of fungi that attack garlic, including *Alternaria porri* (Ellis) Cif. [109], *Fusarium oxysporum* f. sp. cepae [110], *Penicillium hirsutum* [111,112], *Pyrenochaeta terrestris* (H.N.). Although certain white-skinned varieties also show resistance, red-skinned garlic types often show stronger resistance to *Embellisia allii* than white-skinned cultivars [113]. However, there are other accounts of attempts to find resistance that were unsuccessful, such as those with *Sclerotium cepivorum* [114], *Puccinia allii* F. Rudolphi [115], and *Penicillium hirsutum* [116]. The complicated processes and difficulties involved in detecting resistance in germplasm are shown by the divergent findings from numerous studies on the same diseases.

### 5.2. Genetic Modification

Particle bombardment, often referred to as biolistic transformation, can change garlic using plasmid DNA [11,28]. Significant consequences for the transmission of resistance genes in garlic result from this method. Gamma radiation is proposed as a method for inducing beneficial mutations in garlic, potentially leading to disease resistance [117] (Figure 4).

## 6. Challenges

Due to the involvement of several small farmers and gardeners, garlic production is not only of horticultural importance but also has societal value. Numerous cities in North America, Europe, and the UK have annual garlic fairs and festivals. Bulb garlic is frequently exchanged or sold as seed and eaten for culinary purposes. The internet has given garlic aficionados, especially those with modest financial means but plenty of zeal and knowledge, a forum for enhanced communication and germplasm exchange. While there are numerous advantages to this, there are also hazards related to the transfer of infections along with genetic material. Even if the majority of garlic infections are common, there is always a risk of spreading novel diseases or more aggressive genotypes into fields that had not previously been impacted. The early identification of infections in seed garlic has improved because of developments in diagnostic technology. Affordably priced viral detection kits may provide rapid and accurate testing, enabling producers to recognize affected plants and implement the proper disease-control strategies. Additionally, introducing disease-resistant cultivars created via conventional breeding or genetic alteration has the potential to improve disease control in seed-garlic production. For both their personal use and the market, growers with more financial means are progressively able to use tissue-culture programs to produce disease-free planting material. It is predicted that the establishment of cooperatives or other methods would make tissue culture accessible to smaller producers as well. Both big and small farmers can benefit from improvements in diagnostic technologies, including readily available and reasonably priced viral testing and disease-resistant cultivars. 

## Figures and Tables

**Figure 1 plants-13-00267-f001:**
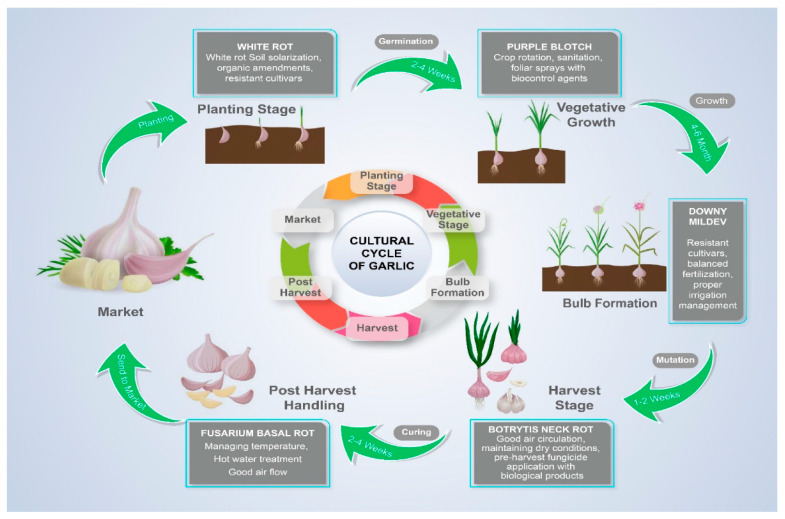
Cultural Cycle of Garlic.

**Figure 2 plants-13-00267-f002:**
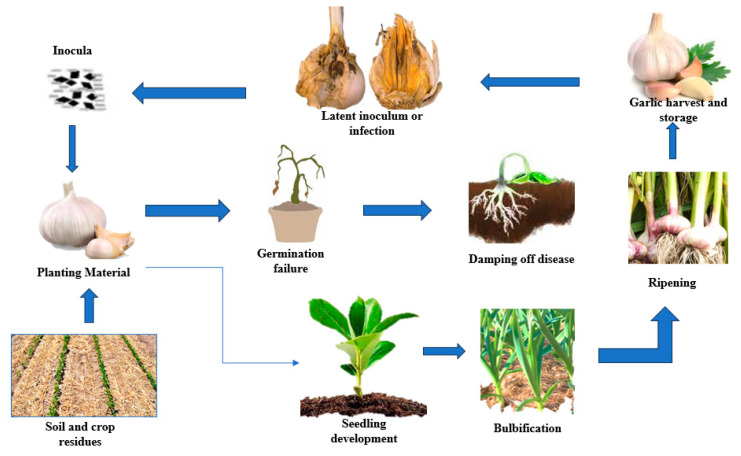
Cycle of fusarium dry rot in garlic.

**Figure 3 plants-13-00267-f003:**
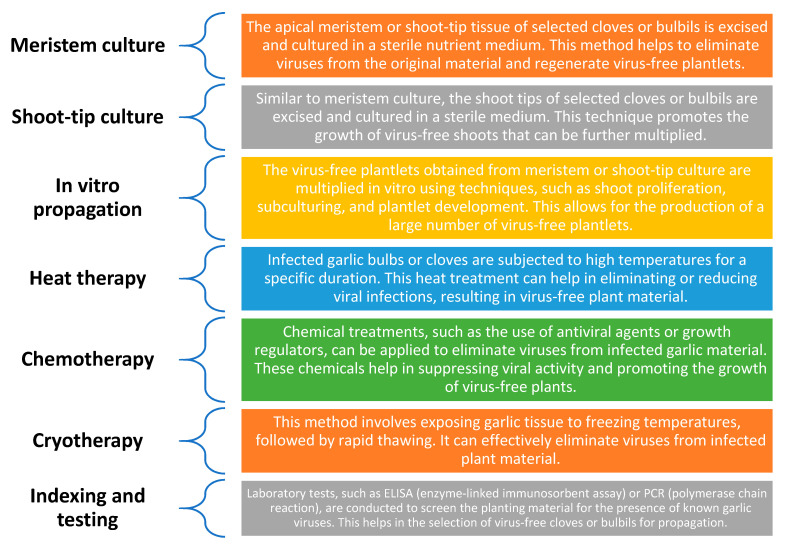
Virus free garlic seed production.

**Figure 4 plants-13-00267-f004:**
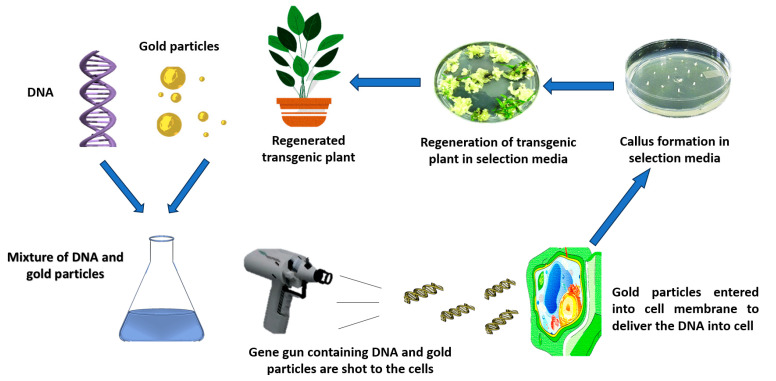
Particle bombardment for genetic transformation.

**Table 1 plants-13-00267-t001:** Worldwide production of garlic.

Rank	Country	Production (Tonnes)	Yield (Tonnes/ha)	% of Global Production	Reference
**1**	China	20,000,000	18	71.40%	[2]
**2**	India	1,250,000	12	4.50%	[7]
**3**	Bangladesh	466,389	15	1.70%	[8]
**4**	South Korea	387,671	17	1.40%	[9]
**5**	Egypt	318,800	13	1.10%	[10]
**6**	Spain	271,350	19	1.00%	[11]
**7**	United States	237,340	16	0.80%	[12]

**Table 2 plants-13-00267-t002:** Fungal pathogens in garlic seed.

Fungal Pathogen	Common Name	Disease Symptoms	References
*Penicillium* spp.	Blue Mold	Decay of seed cloves during storage	[38,39]
*Botrytis* spp.	Gray Mold	Gray mold on garlic bulbs	[40,41]
*Fusarium* spp.	Basal Rot	Basal rot and vascular wilt in plants	[20,42]
*Sclerotium* spp.	White Rot	White rot in garlic bulbs	[43]
*Rhizoctonia* spp.	Root Rot	Damping off, root rot, and basal plate rot	[44,45]
*Alternaria* spp.	Leaf Blight	Leaf blight and bulb rot	[46]
*Pythium* spp.	Damping off	Damping off and root rot in seedlings	[47]
*Sclerotinia* spp.	White Mold	White mold on garlic bulbs	[48]
*Colletotrichum* spp.	Anthracnose	Anthracnose with sunken lesions	[49]
*Myrothecium* spp.	Bulb Rot	Bulb rot and leaf blight	[50]

**Table 3 plants-13-00267-t003:** Bacterial species causing diseases in garlic seed.

Bacterial Pathogen	Common Name	Disease Symptoms	References
*Xanthomonas* spp.	Bacterial Leaf Spot	Water-soaked lesions on leaves and bulbs	[65]
*Pseudomonas* spp.	Soft Rot	Softening and decay of bulbs	[66]
*Erwinia* spp.	Bacterial Bulb Decay	Slimy rotting of bulbs	[67]
*Pantoea* ananatis	Center Rot	Rotting and discoloration of bulb centers	[68]
*Clavibacter* spp.	Bacterial Canker	Raised, corky cankers on leaves and stems	[69]
*Burkholderia cepacia*	Bulb Rot	Rotting and foul odor in bulbs	[70]
*Enterobacter cloacae*	Basal Plate Rot	Rotting at the base of bulbs	[71]
*Dickeya* spp.	Blackleg	Blackened and soft rotting of stems	[72]
*Agrobacterium tumefaciens*	Crown Gall	Tumor-like growths on stems and roots	[73]

Another significant issue in the development of seed garlic is viral infections. The common viruses that affect garlic are the onion yellow dwarf virus (OYDV), leek yellow stripe virus (LYSV), garlic common latent virus (GCLV), and shallot latent virus (SLV). Viral infections can result in slowed development, smaller bulbs, and generally less healthy plants. The efforts to control diseases are made more difficult by the nonpersistent transmission of these viruses by aphids and other vectors.

**Table 4 plants-13-00267-t004:** Viruses affecting garlic seeds and their vectors.

Virus	Common Name	Symptoms and Effects	Vectors	References
Garlic common latent virus (GCLV)	Common Latent Virus	No visible symptoms, latent infection in garlic plants	Unknown	[74]
Garlic mosaic virus (GarMV)	Garlic Mosaic Virus	Mosaic patterns on leaves, stunted growth, reduced yield	Aphids (*Myzus persicae*)	[75]
Leek yellow stripe virus (LYSV)	Leek Yellow Stripe Virus	Yellow stripes on leaves, stunted growth, bulb deformities	Onion thrips (*Thrips tabaci*)	[76]
Shallot latent virus (SLV)	Shallot Latent Virus	No visible symptoms, latent infection in shallots	Unknown	[77]
Onion yellow dwarf virus (OYDV)	Onion Yellow Dwarf Virus	Stunted growth, yellowing of leaves, bulb size reduction	Onion thrips (*Thrips tabaci*)	[78]
Iris yellow spot virus (IYSV)	Iris Yellow Spot Virus	Yellow spots on leaves, necrotic streaks, bulb damage	Onion thrips (*Thrips tabaci*)	[79]
Cucumber mosaic virus (CMV)	Cucumber Mosaic Virus	Mosaic patterns, leaf curling, plant stunting	Aphids (various species)	[80]
Tobacco rattle virus (TRV)	Tobacco Rattle Virus	Stunted growth, yellowing, necrosis, bulb deformities	Soil-borne nematodes (*Trichodorus* spp.)	[81]
Shallot virus X (ShVX)	Shallot Virus X	Yellowing, stunted growth, distorted bulbs	Unknown	[82]
Garlic latent virus (GarLV)	Garlic Latent Virus	No visible symptoms, latent infection in garlic plants	Unknown	[83]

The development of garlic germplasm is disadvantaged by viral infections, both because of the decreased yield and quality and because minor virus signs might be misconstrued for varietal variations in the garlic germplasm. Garlic is commonly infected by a number of viruses, including the onion yellow dwarf virus (OYDV), which is spread by aphids like *Myzus persicae* and a number of other aphid species in a nonpersistent way; leek yellow stripe virus (LYSV), which is also aphid-transmitted in a temporary manner by various aphid species; garlic common latent virus (GCLV), transmitted through mechanical inoculation and aphids; shallot latent virus (SLV), transmitted nonpersistently by *Myzus ascolonicus* (Sciamyzus); and, possibly, *Aphis fabae* [84]. Garlic is infected by the tobacco rattle virus (TRV), which is transmitted by nematodes of the *Trichodoridae* family [85]. A developing issue in onion, leek, and, to a lesser extent, garlic is the iris yellow spot virus (IYSV), which is spread by thrips (*Thrips tabaci*) [86]. There is ambiguity around the names of viruses that cause mosaic symptoms in garlic, such as the garlic mosaic virus.

**Table 5 plants-13-00267-t005:** Nematodes affecting garlic seed.

Nematode Pest	Common Name	Damage Symptoms	References
*Ditylenchus dipsaci*	Stem and Bulb Nematode	Stunted growth, leaf yellowing, bulb rot, and reduced yield	[95]
*Meloidogyne* spp.	Root-Knot Nematode	Galls on roots, stunted growth, nutrient deficiency	[96]
*Pratylenchus* spp.	Lesion Nematode	Lesions on roots, reduced root system, poor nutrient uptake	[97]
*Tylenchulus semipenetrans*	Citrus Nematode	Feeding damage on roots, decline in plant health	[98]
*Heterodera* spp.	Cyst Nematode	Formation of cysts on roots, stunted growth, yield loss	[99]
*Xiphinema* spp.	Dagger Nematode	Feeding damage on roots, yellowing, wilting	[96]
*Longidorus* spp.	Needle Nematode	Stunted growth, root damage, nutrient deficiency	[96]
*Trichodorus* spp.	Sting Nematode	Feeding damage on roots, reduced root system	[100]
*Pratylenchoides* spp.	False Root-Knot Nematode	Root galling, stunted growth, reduced yield	[101]
*Radopholus similis*	Burrowing Nematode	Tunneling in roots, stunting, wilted leaves	[102]

## Data Availability

Data are contained within the article.

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
