# Peer review of "Different Preharvest Diseases in Garlic and Their Eco-Friendly Management Strategies"

_plants, 2024, doi:10.3390/plants13020267_

Round 1

Reviewer 1 Report

Comments and Suggestions for Authors

The review provides interesting information about the different pre-harvest diseases in garlic. The manuscript does provide some information that will be helpful in developing eco-friendly management strategies with additional experiences. Nevertheless, the manuscript is in need of revisions before it is acceptable for publication.

A few points:

Ls.11-12: Revise this sentence to eliminate rewordiness.

Ls.25-26: Repetitive sentence (see line 11).

L.35: Keywords should be in alphabetic order. Also, keywords serve to widen the opportunity to be retrieved from a database. To put words that already are into title and abstracts makes KW not useful. Please choose terms that are neither in the title nor in abstract.

L.40: garlic (Allium sativum L.) production

L.45: Change “Allium sativum L.” by “this plant”

L.66: Again, revise this sentence to eliminate rewordiness.

L.75: Delete “primarily”

L.76: Delete “relatively”

L.86: Delete “significantly”

L.88: …Fusarium, Penicillium, and Alternaria species. These microorganisms can remain…

Ls.120-122: Rephrase this sentence

L.154: Delete “extensively”

Ls.163-165: Summarize this sentence.

L.174: Delete “significantly”

L.202: Delete “effectively”

Ls.260-262: Revise this sentence to eliminate rewordiness.

Comments on the Quality of English Language

No comments.

Author Response

Dear,

Greetings, thank you for reviewing my article. I have made changes according to your comments (track changes). File for point-by-point response to the reviewer’s comments is attached here.

Reviewer 2 Report

Comments and Suggestions for Authors

The review of Anum et alii reports the main phytopathological problems affecting seed garlic production and shows the current possibilities to overcome them.

It is well written and the figures and tables included help the reader to better follow the text.

I would suggest to include in the Introduction also some data concerning the world production of garlic (surface, yields, major countries).

Also, a figure that depicts the coltural cycle of garlic is suggested to facilitate the comprehension of the review.

Comments on the Quality of English Language

Just some minor spelling

Author Response

Dear Reviewers, Thank you very much for taking the time to review this manuscript. Please find the detailed responses below and the corresponding revisions/corrections highlighted/in track changes in the re-submitted files.

Reviewer 3 Report

Comments and Suggestions for Authors

Comments on the Quality of English Language

Author Response

(The authors gave the same response as above.)
